# Semi-Supervised Underwater Object Detection with Image Enhancement Guided by Attribute-based Data Distribution

## Abstract

Semi-supervised underwater object detection aims to improve the performance of detectors on unlabeled underwater images by leveraging knowledge from labeled ones. However, existing methods often overlook the distribution differences between labeled and unlabeled underwater images. In this paper, we propose a novel underwater image enhancement method guided by attribute-based data distribution (UIEG+), which focuses on reducing the discrepancies between enhanced and original unlabeled images across different attributes, thereby effectively addressing the challenges in semi-supervised underwater object detection. Specifically, we explore an underwater image enhancement strategy based on two attributes: color and scale distributions. For the color attribute, we construct a 3-dimensional grid memory, where each grid cell represents a color subspace and records the number of samples in that subspace. Similarly, for the scale attribute, we design a 1-dimensional vector memory that dynamically stores the number of samples in each scale subspace. Subsequently, we propose an effective sampling method to derive parameters for color and scale transformations based on the aforementioned distribution analysis, increasing the likelihood of transformations in low-distribution regions. To evaluate its effetiveness and superiority, massive semi-superivised underwater object deteciton experiments in multiple datasets have been conduted by integrating UIEG+ into existing semi-supervised object detection frameworks. The code will be released.

## 1 Introduction

With the development of deep learning, object detection models have also experienced unprecedented growth, and many outstanding detection models have been proposed (Ren et al., 2017; Tian et al., 2019; Carion et al., 2020). However, when dealing with underwater scenes and objects, these detectors still face challenges related to complexity and diversity. To address these issues, a simple solution is to gather a more extensive dataset that encompasses a wide range of scenes and object styles. Nevertheless, this is an extremely challenging task that requires significant resources and makes it difficult to cover all possible scenarios.

In recent years, semi-supervised object detection based on image enhancement has also emerged as a pivotal solution, which primarily emphasizes the generation of high-quality enhanced images accompanied by labels or pseudo-labels. Existing image enhancement methodologies for semi-supervised underwater object detection (SSUOD) can be broadly classified into two categories. The former encompasses traditional image enhancement techniques (Zhang et al., 2022; Peng et al., 2018; Liu et al., 2023b), which employ physics-based operations to manipulate various attributes such as color, scale, and contrast, often through random transformations, as illustrated in Fig. 1 (a). Despite their widespread application in semi-supervised object detection frameworks, such as the teacher-student model with weak and strong augmentations (Zhang et al., 2023b), these methods exhibit certain limitations: firstly, they fail to account for the distribution of unlabeled images across different attribute spaces, potentially resulting in the generation of unrealistic enhanced images that could detrimentally affect model training; secondly, they overlook the adverse effects of the imbalanced distribution of unlabeled images within these attribute spaces. The latter involves the construction of learnable networks (Jiang et al., 2022; Cong et al., 2023; Zhang et al., 2024) to

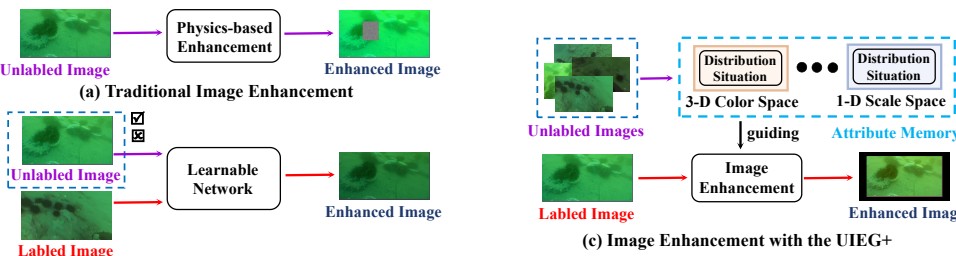

Figure 1: Comparison between the proposed UIEG+ and other methods. (a) Traditional image enhancement; (b) Image enhancement with learnable network; (c) Our UIEG+ method. The check mark and cross in the rectangular boxes indicate whether the unlabeled image is used or not.

achieve style transfer between labeled and unlabeled images. The works (Peng et al., 2023; Zhang et al., 2024) focus on designing end-to-end networks that directly facilitate style transfer across different domains. In contrast, other studies (Deng et al., 2022; Wang et al., 2023b) extract style information and utilize it to guide the content image through style transformation. However, these approaches also present limitations when applied to semi-supervised underwater detection: firstly, they are constrained to enhancing images within the three-dimensional color space, neglecting potential transformations in other attribute spaces, such as scale space; secondly, these methods are similarly challenged by the issue of imbalanced distribution previously mentioned.

**Contribution** Different from the aforementioned methodologies, this paper considers the actual distribution of unlabeled underwater images, which aims to mitigate the distributional discrepancies between enhanced and unlabeled underwater images. To be specific, we address the distribution of unlabeled underwater images in terms of color and scale attributes. For the color attribute, we develop a three-dimensional color memory, wherein each cell represents a three-dimensional color subspace within a defined range and records the number of unlabeled images in that subspace. Analogously, for the scale attribute, we construct a one-dimensional memory that records the number of unlabeled images in each cell. The color and scale subspaces of the images are determined using the mean values of each color channel and the average scale of all objects within the image. Subsequently, based on these distributional insights, we propose a sampling strategy to derive color and scale transformation parameters, thereby guiding image enhancement. This sampling method effectively balances enhanced images across different subspaces of color and scale. Ultimately, we integrate the proposed UIEG+ into existing SSOD frameworks to address semi-supervised underwater object detection, as shown in Fig. 1.

In summary, our contributions can be outlined as follows:

- We propose a novel underwater image enhancement method guided by attribute-based data distribution (UIEG+), which aims to reduce distributional differences between enhanced and unlabeled underwater images by analyzing the distribution of unlabeled images in terms of color and scale attributes.

- We incorporate the proposed UIEG+ into existing SSOD frameworks, thereby effectively addressing the challenges of semi-supervised underwater object detection.

- Our method achieves state-of-the-art performance on multiple semi-supervised underwater detection (SSUD) datasets and demonstrates significant improvements across various SSOD methods.

## 2 RELATED WORK

### 2.1 OBJECT DETECTION

Deep learning detectors have made significant advancements over the past few years. Currently, existing detection models are generally classified into two categories, ı.e one- and two-stage detectors. One-stage detectors (*e.g.*FCOS (Tian et al., 2019), Yolo (Redmon & Farhadi, 2018), and

DETR (Carion et al., 2020)) directly predict the classes and rectangular boxes of objects on feature maps without medium proposals. Different from one-stage detectors, two-stage detectors (Ren et al., 2017; Cai & Vasconcelos, 2018) design the region proposal network (RPN) to obtain more accurate proposals and further conduct the classification and regression of these proposals. In this work, we explain the principles of our method based on the PseCo framework (Li et al., 2022) with Faster-RCNN (Ren et al., 2017) detector and demonstrate its effectiveness and generalizability when applied to different detectors (Tian et al., 2019) in our experiments.

## 2.2 SEMI-SUPERVISED DETECTION

Semi-supervised detection seeks to improve the performance of detectors on unlabeled images by exploring the training strategies on labeled and unlabeled images. One branch (Zhou et al., 2022; Liu et al., 2023a; Shehzadi et al., 2024) performs self-training on labeled and unlabeled images. The core idea is to acquire high-quality pseudo-labels or samples. Some works (Wang et al., 2021; Li et al., 2023) pay attention to uncertain regions or class imbalance problems. Despite their effectiveness, these methods are limited by the noise in predicted pseudo-labels or samples, which primarily arises from differences between labeled and unlabeled images and the selection of hyperparameters. Moreover, another popular trend (Jeong et al., 2019; Li et al., 2022; Wang et al., 2023a) is to employ consistency analysis of internal features or external predictions from the detection network to learn more robust feature representations or outputs. However, achieving more effective consistency guidance is challenging, particularly when it comes to the feature representation of foreground objects affected by various disturbances. In this work, we utilize our UIEG+ method instead of the strong augmentation found in previous teacher-student frameworks (Zhou et al., 2022; Li et al., 2022; Liu et al., 2023a), resulting in enhanced images that match the data distribution of the unlabeled images.

## 2.3 SEMI-SUPERVISED UNDERWATER DETECTION

Semi-supervised underwater detection aims at improving the adaptation of detectors to various underwater scenes by leveraging both labeled and unlabeled images in the training process. Existing works (Sharma et al., 2024; Noman et al., 2021) mainly address this issue by using current semi-supervised detection strategies. Different from these works, we propose an underwater image enhancement guided by attribute-based data distribution to reduce the distribution discrepancies between enhanced images and the original unlabeled images in different attributes, effectively handling semi-supervised underwater object detection.

## 2.4 IMAGE EHANCEMENT

Image enhancement is an effective method to handle semi-supervised underwater detection. The previous works can be grouped into traditional underwater image enhancement (Zhang et al., 2022; Peng et al., 2018; Liu et al., 2023b) and learning-based underwater image enhancement (Jiang et al., 2022; Cong et al., 2023; Zhang et al., 2024). Traditional underwater image enhancement relies on physics-based operations that are manually defined for different attributes, *i.e.*color (Ancuti et al., 2012; Wang et al., 2018), contrast (Zhang et al., 2022), and scale. These works use random sampling to enhance images but do not account for the distribution of unlabeled images across different attributes in semi-supervised underwater detection. As a result, some unreasonable enhanced images can negatively impact the model's training. In contrast, learning-based underwater image enhancement achieves the style transfer between different domain images by constructing the learnable network. While this approach can reduce style differences between enhanced and unlabeled images, it fails to address issues related to imbalanced style distribution in the unlabeled underwater images and differences in the distribution of other attributes (e.g., scale).

## 2.5 DIFFERENCES FROM OTHER WORKS

In this work, we propose an underwater image enhancement method guided by attribute-based data distribution and integrate it into existing semi-supervised detection frameworks. Compared to the above image enhancement methods, our approach has the following differences. **First**, we consider the distribution of unlabeled underwater images in terms of color and scale attributes, and mitigate the impact of imbalanced distributions on enhanced images through a distribution-based sampling

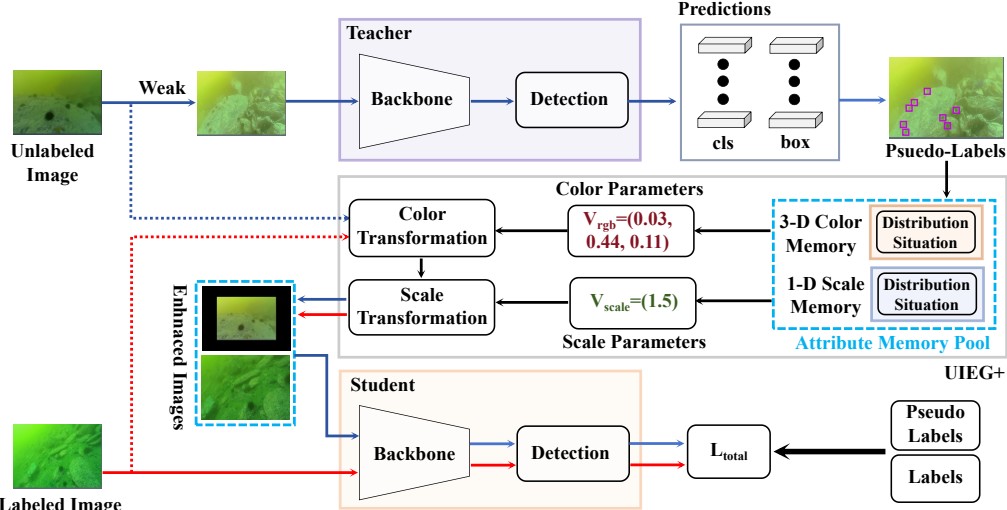

Figure 2: The framework of UIEG+-based semi-supervised underwater object detection, which is built upon the PseCo (Li et al., 2022) with Faster-RCNN (Ren et al., 2017) detector. Compared to the original PseCo, we replace the strong augmentation with our UIEG+ method.

method. **Second**, our method effectively enhances both labeled and unlabeled images without requiring additional training. In summary, our UIEG+ is a simple yet effective approach that boosts the performance of existing semi-supervised detection frameworks for underwater detection.

## 3 METHODOLOGY

### 3.1 OVERALL SSUOD FRAMEWORK

Fig. 2 illustrates the framework of semi-supervised underwater object detection with UIEG+, which is constructed based on the PseCo (Li et al., 2022) with Faster-RCNN (Ren et al., 2017) detector and the proposed UIEG+. Given the labeled image $(x_l, y_l)$ and the unlabeled image $x_u$, we first perform weak augmentation on the unlabeled underwater image and generate its pseudo-labels $\hat{y}_u$ using the teacher detector. Then, we update the 3-D color and 1-D scale memories with the unlabeled underwater image and its pseudo-labels. Following this, we sample transformation parameters for color and scale from these memories and apply them to both labeled and unlabeled underwater images, producing enhanced images with corresponding labels and pseudo-labels, i.e, $(x_l^{ie}, y_l^{ie})$ and $(x_u^{ie}, \hat{y}_u^{ie})$. Finally, the student detector is trained using these labeled, unlabeled and enhanced images. Therefore, the unified optimization loss is mathematically described as follows:

$$\mathcal{L}_{\text{total}} = \mathcal{L}_{det}(x_l, y_l) + \gamma_l^{ie}\mathcal{L}_{det}(x_l^{ie}), y_l^{ie}) \\ + \gamma_u^{ie}\mathcal{L}_{det}(x_u^{ie}), \hat{y}_u^{ie}) \tag{1}$$

where $\gamma_l^{ie}$ and $\gamma_u^{ie}$ are weight coefficients used to balance three loss terms, and $\mathcal{L}_{det}$ denotes the detection loss of the base student detector Faster-RCNN (Ren et al., 2017).

Notably, our UIEG+ can be applied to various semi-supervised object detection frameworks (*e.g.*, DenseTeacher (Zhou et al., 2022) and ARSL (Liu et al., 2023a)). In this work, we take PseCo (Li et al., 2022) with Faster-RCNN as an example to illustrate the principles of UIEG+ and explain how to integrate it into PseCo. In addition, we demonstrate the generality of our UIEG+ by applying it to various SSOD frameworks in the experiments.

### 3.2 UNDERWATER IMAGE ENHANCEMENT GUIDED BY ATTRIBUTE-BASED DATA DISTRIBUTION

In this section, we propose an underwater image enhancement method guided by attribute-based data distribution to minimize the distribution differences between enhanced and unlabeled underwater

Figure 3: (a) 3-D color memory; (b) 1-D scale memory; (c) The Process of storing scale information; (d) The process of image enhancement in scale space.

images. This approach effectively improves the relevance and diversity of the enhanced images. As shown in Fig. 2, to end this, we first create an attribute memory pool consisting of the 3-D color memory and 1-D scale memory. Subsequently, we analyze unlabeled images with pseudo-labels in color and scale spaces separately, updating the corresponding memory cells by assigning them to different subspaces. Following this, we proceed to enhance the images based on the distribution of the unlabeled images within these memory pools.

**Color and Scale Memories** Color and scale memories are used to record the number of unlabeled underwater images in different subspaces. Specifically, we divide the 3-dimensional color space, with each dimension ranging from 0 to 1, into $(\lfloor 1/d_{rgb} \rfloor + 1)^3$ subspaces using the interval step $d_{rgb}$ in each dimension. Here, $\lfloor \cdot \rfloor$ is the floor operation, and $(\lfloor 1/d_{rgb} \rfloor + 1)$ is labeled as $N_{rbg}$. Following this, we construct a 3-D color memory $M_{rgb} \in R^{N_{rgb} \times N_{rgb} \times N_{rgb}}$, where each cell corresponds to a subspace with the size of $d_{rgb} \times d_{rgb} \times d_{rgb}$. Similarly, we divide the scale space, ranging from 0 to $\infty$, into $\lfloor 1/d_{scale} \rfloor + 1$ subspaces using the interval step $d_{scale}$. Meanwhile, the scale memory is constructed, where each cell corresponds to a subspace with the size of $d_{scale}$. The structure of color and scale memories is shown in Fig. 3 (a) and (b).

**Sample Distribution** Following the establishment of the color and scale memories, we analyze the distribution of unlabeled underwater images. Given unlabeled image $x_u \in R^{3 \times H \times W}$, we normalize it to the range of $[0, 1]$ and then compute the mean value of $x_u$ for each dimension, denoted as $u_{rgb} \in R^3$.

$$u_{rgb}^k = \frac{1}{HW} \sum_{i,j}^{H,W} x_u^{k,i,j} \tag{2}$$

where $H$ and $W$ are the height and width of the unlabeled underwater image. $k$ is the number of dimensions of the image. Then, we further allocate the image $x_u$ into the color memory space using $u_{rgb} \in R^3$ to determine the corresponding position index $e_{rgb} \in Z^3$. By utilizing the index $e_{rgb}$, the color memory $M_{rgb}$ is updated as follows.

$$e_{rgb}^k = \lfloor u_{rgb}^k / d_{rgb} \rfloor \tag{3}$$

$$M_{rgb}(e_{rgb}) = M_{rgb}(e_{rgb}) + 1 \tag{4}$$

Similarly, the scale memory is updated by determining the location index of the unlabeled underwater image in it, which can be formulated as:

$$u_{scale} = \frac{1}{N_{box}} \sum_{i}^{N_{box}} \sqrt{(B_{i,2} - B_{i,0}) \cdot (B_{i,3} - B_{i,1})} \tag{5}$$

$$M_{scale}(e_{scale}) = M_{scale}(e_{scale}) + 1, \quad e_{scale} = \lfloor u_{scale} / d_{scale} \rfloor \tag{6}$$

where $B \in R^{N_{box} \times 4}$ represents the bounding boxes of all objects in the unlabeled underwater image, obtained by the teacher detector. The process of analyzing the distribution of unlabeled underwater images in the scale space is illustrated in Fig 3 (c).

**Image Enhancement** To reduce the distribution differences between enhanced and unlabeled images, in this section, we explore the image enhancement method based on the distributions of unlabeled underwater images in color and scale space. In detail, we first perform the color transformation for input images. As shown in Fig. 2, we utilize the distribution information in the color memory to determine the sampling probability of each cell within it, which can be formulated as

$$P_{rgb} = (1 - \frac{M_{rgb}}{max(M_{rgb})}) \cdot U_{3d} \cdot I_{rgb} \tag{7}$$

$$I_{rgb} = \begin{cases} 1.0, M_{rgb} > 0 \\ 0.0, otherwise \end{cases} \tag{8}$$

where $U_{3d}$ is a 3-dimensional matrix obtained by randomly sampling from a uniform distribution in the range $[0, 1]$, which has the same size as $M$. The color transformation parameter is then determined based on the sampling probability, which can be formulated as

$$e^*_{rgb} = argmax(P_{rgb}) \tag{9}$$

$$V_{rgb} = (e^*_{rgb} + \sigma_{rgb}) \cdot d_{rgb} \tag{10}$$

where $\sigma_{rgb}$ is a contrast bias less than 1. $argmax(\cdot)$ is the function that finds the index of the maximum value. $V_{rbg}$ is the final parameter of the color transformation. By employing the parameter $V_{rbg}$, we perform the color transformation for input images, which can be formulated as

$$x^{ie} = x + \eta_{rgb} \cdot (V_{rbg} - u^x_{rgb}) \tag{11}$$

where $\eta_{rgb}$ is a constant and $u^x_{rgb}$ is the mean value of $x$ in each dimension. Following this, we further derive the scale transformation parameter using a process similar to that used for the color transformation, which can be formulated as

$$P_{scale} = (1 - \frac{M_{scale}}{max(M_{scale})}) \cdot U_{1d} \cdot I_{scale}, \quad I_{scale} = \begin{cases} 1.0, M_{scale} > 0 \\ 0.0, otherwise \end{cases} \tag{12}$$

$$V_{scale} = \frac{(e^*_{scale} + \sigma_{scale}) \cdot d_{scale}}{u^x_{scale}}, \quad e^*_{scale} = argmax(P_{scale}) \tag{13}$$

where $U_{1d}$ is a 1-dimensional matrix and $\sigma_{scale}$ is a contrast bias less than 1. $u^x_{scale}$ is the mean scale value of all objects in the image $x$. $V_{scale}$ is the scale rate of the image $x$. By applying $V_{scale}$ to the interpolate operation (Jaderberg et al., 2015), we achieve the scale transformation of $x$. Fig. 3 (d) shows the process of enhancing the image in terms of scale.

Finally, in this work, we enhance the images of both unlabeled and labeled datasets using the proposed UIEG+ to increase the rationality and diversity of the enhanced images, effectively addressing the problem of semi-supervised underwater object detection with Eq. 1.

### 3.3 OPTIMIZATION

The training process of our UIEG+-based SSU detection framework is meticulously structured into two distinct stages to ensure optimal performance. In Stage 1, we concentrate exclusively on training the teacher detector using only the labeled images. In Stage 2, we focus on optimizing the student detector training on both labeled and unlabeled images. The objective here is to leverage the vast amount of unlabeled data to enhance the model's generalization capabilities. With the loss function in Eq. 1 to guide the learning process of the student detector, we can continuously update the teacher detector using the exponential moving average (EMA) strategy to ensure that it evolves alongside the student detector, which can be expressed as:

$$\theta_t = \delta \cdot \theta_t + (1 - \delta) \cdot \theta_s \tag{14}$$

where $\theta_t$ and $\theta_s$ indicate the parameters of the teacher and student detectors. $\delta$ represents the EMA rate.

Table 1: Results of our approach and comparison to state-of-the-arts on DUO. The best two results are shown in red and blue fonts, respectively.

| Method | Detector | Holothurian | Echinus | Scallop | Starfish | $mAP_{50}$ | mAP |
|---|---|---|---|---|---|---|---|
| Labeled Only | FCOS | 66.8 | 86.5 | 21.0 | 83.6 | 64.5 | 39.2 |
| DenseTeacher (Zhou et al., 2022) | FCOS | 77.7 | 88.7 | 35.2 | 87.9 | 72.4 | 48.9 |
| ARSL (Liu et al., 2023a) | FCOS | 78.1 | 85.2 | 42.9 | 88.5 | 73.7 | 35.4 |
| Unbiased-Teacherv2 (Liu et al., 2022) | FCOS | 82.4 | 86.8 | 50.7 | 88.8 | 77.2 | 56.8 |
| Consistent-Teacher (Wang et al., 2023a) | FRCNN | 84.1 | 87.6 | 52.3 | 86.8 | 77.7 | 60.0 |
| Labeled Only | FRCNN | 66.1 | 85.6 | 16.8 | 84.3 | 63.2 | 40.4 |
| Soft Teacher (Xu et al., 2021) | FRCNN | 84.9 | 88.2 | 44.3 | 89.2 | 76.6 | 53.9 |
| PseCo (Li et al., 2022) | FRCNN | 85.7 | 88.5 | 48.7 | 89.3 | 78.0 | 56.8 |
| MixTeacher (Liu et al., 2023c) | FRCNN | 85.4 | 88.9 | 46.6 | 89.6 | 77.6 | 56.3 |
| Ours (DenseTeacher) | FCOS | 79.1 | 88.4 | 41.1 | 87.3 | 74.0 | 50.9 |
| Ours (ARSL) | FCOS | 78.8 | 85.5 | 45.4 | 88.5 | 74.6 | 36.1 |
| Ours (PseCo) | FRCNN | 85.9 | 89.4 | 54.3 | 90.3 | 80.0 | 57.2 |
| oracle | FCOS | 80.9 | 89.3 | 40.7 | 88.1 | 74.8 | 53.6 |
| oracle | FRCNN | 87.0 | 90.7 | 61.9 | 91.3 | 82.7 | 62.6 |

# 4 EXPERIMENTS

## 4.1 IMPLEMENTATION

Our framework is implemented based on the PseCo framework (Li et al., 2022) in PyTorch (Paszke et al., 2019). Similarly, we use Faster-RCNN (Ren et al., 2017) with ResNet-50 (He et al., 2016) pre-trained on ImageNet (Krizhevsky et al., 2012) as teacher and student detectors. The model is trained with a learning rate of 1e-3, momentum of 0.9, and weight decay of 1e-4. $\gamma_l^{ie}$ and $\gamma_u^{ie}$ are set to 1.0 and 0.1 in Eq. 1, respectively. $d_{rgb}$ and $d_{scale}$ are set to 0.05 and 64, respectively. $\eta_{rgb}$ is 0.3 in Eq. 11. To evaluate the effectiveness and generalizability of our UIEG+, we integrate it into two additional SSOD frameworks, *i.e.* ARSL (Liu et al., 2023a) and DenseTeacher (Zhou et al., 2022), utilizing both FCOS (Tian et al., 2019) as the base detectors. Specifically, in both ARSL and DenseTeacher, we substitute only the strong augmentation with our UIEG+, keeping all other components unchanged.

## 4.2 DATASETS

In this work, we conducted experiments on two underwater datasets to verify the effectiveness and superiority of the proposed UIEG+. For all experiments, we take the standard mean average precision (mAP) at differrent IoU thresholds (*e.g.*, AP$_{50:95}$ denoted as mAP, AP$_{50}$) as our evaluation metrics.

- **DUO** The DUO (Liu et al., 2021) dataset has 6671 images for training and 1111 images for evaluation with shared four classes, ι.e, Holothurian, Echinus, Scallop, and Starfish. Following works (Li et al., 2022; Hua et al., 2023), we randomly sample 10% of the training images as labeled data, while the remaining images are used as unlabeled data. The testing images are used for evaluation.
- **URPC** The URPC dataset, collected from the Underwater Robot Picking Contest, includes 16247 training images and 4062 testing images, with four classes ι.e, Holothurian, Echinus, Scallop, and Starfish. Similarly, we randomly sample 10% of the training images as labeled data, leaving the remaining images as unlabeled data. Additionally, the testing images are used for evaluation.

## 4.3 RESULT ANALYSIS

In this section, we present the evaluation results of current state-of-the-art SSOD methods and demonstrate the effectiveness of our UIEG+ by applying it to other SSOD frameworks on three sets of experiments built on two underwater datasets, *e.g.* DUO and URPC. The "Labeled Only"

Table 2: Results of our approach and comparison to state-of-the-arts on URPC. The best two results are shown in red and blue fonts, respectively.

| Method | Detector | Holothurian | Echinus | Scallop | Starfish | $AP_{50}$ | mAP |
|---|---|---|---|---|---|---|---|
| Labeled Only | FCOS | 64.6 | 74.1 | 66.8 | 70.2 | 68.9 | 30.2 |
| DenseTeacher (Zhou et al., 2022) | FCOS | 74.5 | 89.6 | 73.2 | 81.3 | 79.7 | 41.9 |
| ARSL (Liu et al., 2023a) | FCOS | 75.0 | 88.0 | 73.8 | 80.8 | 79.4 | 34.3 |
| Unbiased-Teacherv2 (Liu et al., 2022) | FCOS | 70.8 | 80.6 | 63.9 | 75.7 | 72.8 | 38.9 |
| Consistent-Teacher (Wang et al., 2023a) | FCOS | 79.2 | 84.2 | 72.3 | 78.9 | 78.7 | 43.3 |
| Labeled Only | FRCNN | 69.5 | 87.6 | 70.9 | 79.5 | 76.9 | 39.3 |
| Soft Teacher (Xu et al., 2021) | FRCNN | 78.3 | 88.2 | 72.3 | 79.4 | 79.5 | 43.2 |
| PseCo (Li et al., 2022) | FRCNN | 80.6 | 87.3 | 74.7 | 79.9 | 80.6 | 44.3 |
| MixTeacher (Liu et al., 2023c) | FRCNN | 79.5 | 87.9 | 73.9 | 79.1 | 80.1 | 43.8 |
| Ours (DenseTeacher) | FCOS | 77.4 | 89.5 | 74.8 | 82.4 | 81.0 | 43.9 |
| Ours (ARSL) | FCOS | 76.3 | 88.7 | 74.3 | 81.3 | 80.2 | 34.7 |
| Ours (PseCo) | FRCNN | 80.6 | 89.8 | 76.2 | 81.8 | 82.1 | 44.8 |
| oracle | FCOS | 86.2 | 92.3 | 80.5 | 86.4 | 86.3 | 50.0 |
| oracle | FRCNN | 88.0 | 92.6 | 82.3 | 86.6 | 87.4 | 51.7 |

Table 3: Results of our approach and comparison to state-of-the-arts on DUO and URPC. The best two results are shown in red and blue fonts, respectively.

| Method | Detector | Holothurian | Echinus | Scallop | Starfish | $AP_{50}$ | mAP |
|---|---|---|---|---|---|---|---|
| Unbiased-Teacherv2 (Liu et al., 2022) | FCOS | 62.8 | 78.2 | 48.0 | 74.2 | 65.8 | 33.4 |
| ARSL (Liu et al., 2023a) | FCOS | 66.0 | 83.2 | 56.6 | 80.1 | 71.5 | 29.8 |
| PseCo (Li et al., 2022) | FRCNN | 72.5 | 85.0 | 51.1 | 79.6 | 72.0 | 35.0 |
| MixTeacher (Liu et al., 2023c) | FRCNN | 68.4 | 85.2 | 47.0 | 79.8 | 70.1 | 34.2 |
| Ours (ARSL) | FCOS | 66.6 | 84.6 | 63.5 | 79.3 | 73.5 | 30.5 |
| Ours (PseCo) | FRCNN | 71.5 | 83.7 | 63.9 | 73.7 | 73.2 | 35.5 |

results indicate that the model is trained only on labeled images and then evaluated on testing images. The "oracle" results indicate the model is trained on all training images (including labeled and unlabeled images) and evaluated on testing images.

**DUO** Tab. 1 exhibits the evaluation results on DUO dataset. As shown in Tab. 1, when the proposed UIEG+ is applied to DenseTeacher, ARSL, and PseCo, our approach improves $AP_{50}$ from 72.4%, 73.7% and 78.0% to 74.0%, 74.6% and 80.0%, resulting in gains of 1.6%, 0.9% and 2.0%. Additionally, our method achieves mAP improvements of 2.0%, 0.7% and 0.4%, respectively. Compared to the baseline (Labeled only) on Faster R-CNN and FCOS detectors, our UIEG+ demonstrates a significant improvement, further proving the effectiveness of our approach.

**URPC** We display the comparison results on URPC in Tab. 2. As shown in Tab. 2, when strong augmentation in DenseTeacher , ARSL and PseCo is replaced by our UIEG+, they obtain $AP_{50}$ scores of 81.0%, 80.2%, and 82.1%, respectively, achieving clear improvement with 1.3%, 0.8% and 1.5% gains. Meanwhile, our method also acquire mAP gains of 2.0%, 0.4% and 0.5% over DenseTeacher, ARSL, and PseCo, respectively. Additionally, compared to the baseline (Labeled Only), we achieve AP50 gains of 12.1% (81.0% vs. 68.9%) on the FCOS detector, and AP50 gains of 5.2% (82.1% vs. 76.9%) on the Faster R-CNN detector, showing the effectiveness of our UIEG+.

**DUO-to-URPC** To further evaluate our UIEG+ method, we report the comparison results on DUO and URPC in Tab. 3. In this experiment, we use 10% of the training images from DUO as labeled data and 90% of the training images from URPC as unlabeled data. Additionally, the testing images from URPC are utilized for evaluation. As shown in Tab. 3, when our UIEG+ is employed to SSOD frameworks of ARSL and PseCo, these frameworks obtain $AP_{50}$ gains of 2.0% ( 71.5% vs 73.5%) and 1.2% ( 72.0% vs 73.2%), respectively. Moreover, our method also acquire mAP gains of 0.7%, and 0.5% on ARSL and PseCo, verifying the effectiveness of our method.

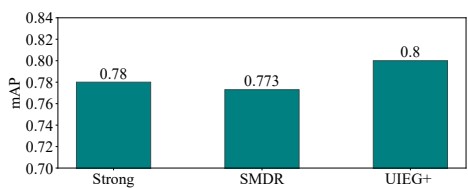

Figure 4: Comparison between image enhancement methods.

Table 4: Effectiveness of important components on URPC.

| CTransfor | STransfor | $AP_{50}$ |
|-----------|-----------|-----------|
|           |           | 73.1      |
| ✓         |           | 77.6      |
| ✓         | ✓         | 80.0      |

## 4.4 ABLATION STUDY

To study the effectiveness of different components and analyze important weights in our UIEG+, we conduct an ablation study based on PseCo framework on DUO dataset.

**Effectiveness of different components** To analyze the effectiveness of image transformations with different attributes (*e.g.*, color and scale) in our UIEG+, we report comparison results of our UIEG+ with various attributes, as shown in Tab. 4. Here, "CTransfor" and "STransfor" represent color and scale transformations, respectively. From Tab. 4, we find that UIEG+ with color transformation significantly improves the performance of PseCo from 73.1% $AP_{50}$ to 77.6% $AP_{50}$. When employing UIEG+ with all two transformations, we obtain the best $AP_{50}$ of 80.0%, evidencing the effectiveness of color and scale transformations in UIEG+.

**Comparison of sampling methods** As shown in Tab. 5, we compare the proposed distribution-based sampling and random sampling for color and scale transformation parameters in UIEG+. Random sampling omits the front part of $U \cdot I$ in Eq. 7 and 12 . From Tab .5, we find that our distribution-based sampling achieves a sig-

Table 5: Comparison of different sampling methods of transformation parameters.

|      | Random | Ours |
|------|--------|------|
| mAP  | 78.7   | 80.0 |

nificant gain of 1.3% compared to random sampling, demonstrating the superiority of our approach.

**Comparison of image enhancement methods** To verify the superiority of our UIEG+ than other image enhancement methods, *e.g.*, randomly strong augmentation and SMDR (Zhang et al., 2023a) that is an image enhancement network, we show comparison results in Fig .4. From Fig .4, we observe that our UIEG+ achieves $AP_{50}$ gains of 2.0% and about 2.3% compared to randomly applied strong augmentation and SMDR.

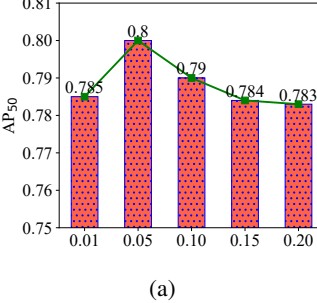
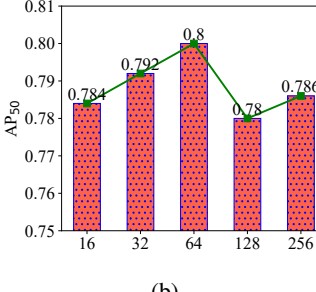
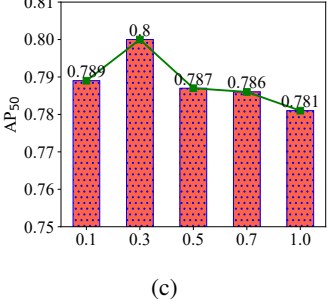

(a)  (b)  (c)

Figure 5: Weight analysis. (a) represents analysis results of $d_{rgb}$; (b) represents analysis results of $d_{scale}$; (c) represents analysis results of $\eta_{rgb}$.

**Analysis on hyperparameter** To explore the effects of weights $d_{rgb}$ in Eq. 3, $d_{scale}$ in Eq. 6 and $\eta_{rgb}$ in Eq. 11, we conduct ablations in Tab .5a, 5b and 5c. As shown in Tab .5a and 5b, both $d_{rgb} = 0.05$ and $d_{scale} = 64$ achieve the best $AP_{50}$ score of 80.0%, respectively. Larger and smaller values of $d_{rgb}$ and $d_{scale}$ can reduce the effectiveness of our UIEG+. In addition, from Tab. 5c, we observe that $\eta_{rgb} = 0.3$ obtains the best $AP_{50}$ score.

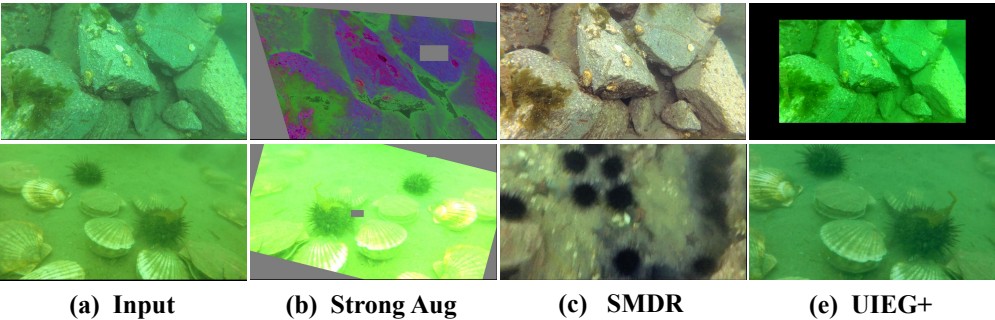

| (a) Input | (b) Strong Aug | (c) SMDR | (e) UIEG+ |

Figure 6: Visual comparisons of our image enhancement method with other methods. The first row of images is from DUO, and the last row is from URPC.

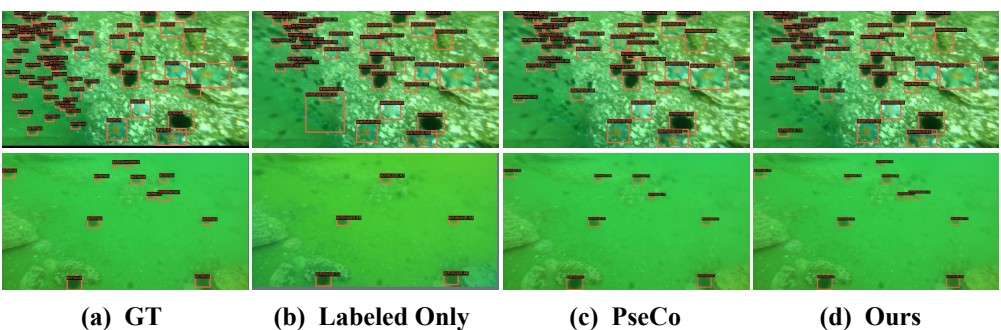

| (a) GT | (b) Labeled Only | (c) PseCo | (d) Ours |

Figure 7: Visual results of PseCo trained with different underwater image enhancement methods on DUO and URPC. (a) shows the ground truth; (b) displays the visual results from the detector trained exclusively on labeled images; (c) presents the results obtained through the PseCo method; (d) illustrates the visual results using our UIEG+. The first row of images is from DUO, and the last row is from URPC.

### 4.5 VISUAL ANALYSIS

To provide a clearer understanding of the effectiveness of our image enhancement method and its effect on improving SSOD framework performance, we show some visualization samples in Fig. 6 and 7.

**Visual comparisons of image enhancement methods** Fig. 6 shows visual comparisons of our image enhancement method with other methods (*e.g.*, strong augmentation and SMDR). From Fig. 6, we can see that strong augmentation can generate unrealistic enhanced images in contrast to the original unlabeled images, negatively impacting detector performance during training. However, our UIEG+ method produces more realistic enhanced images by aligning with the distribution of unlabeled underwater images in terms of color and scale attributes.

**Visual detection results** Fig. 7 provide some visual results of PseCo trained using different image enhancement methods on DUO and URPC datasets. It demonstrates that our UIEG+ can significantly enhance the performance of PseCo for semi-supervised underwater object detection.

## 5 CONCLUSIONS

In this paper, we propose a novel underwater image enhancement method guided by attribute-based data distribution (UIEG+) from a novel perspective, which reduces the differences in color and scale spaces between enhanced and unlabeled images. More importantly, UIEG+ can be flexibly integrated into various SSOD frameworks with different detectors, such as DenseTeacher, ARSL, PseCo, etc. Extensive experimental reuslt can validate the effectiveness and generality of the proposed UIEG+ across multiple benchmarks.

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
