# OpenReview forum: "Semi-Supervised Underwater Object Detection with Image Enhancement Guided by Attribute-based Data Distribution"
_ICLR.cc/2025/Conference — ICLR 2025 Conference Withdrawn Submission_

### Official Review · Reviewer_ppxz · 2024-10-25

**Soundness:** 2
**Presentation:** 2
**Contribution:** 1
**Rating:** 3
**Confidence:** 5

**Summary:**

This paper proposed a dataset-conditioned enhancement for underwater semi-supervised learning.

**Strengths:**

The writing is clear.

**Weaknesses:**

The main weakness is that the novelty is very limited. This paper proposed a dataset-conditioned enhancement for underwater semi-supervised object detection. However, the solution is calculating the average colour/scale parameter from images and use it to augment training data. It may improve the performance, but it's trying to overfit the dataset. As the evaluation dataset is not a big dataset, overfitting may slightly improve the performance. The authors should prove the generalisation of this method. What if the evaluation data contains many objects in different colours and scales? Real underwater scenes are highly diverse, and the URPC data from the Zhangzi Island is heavily biased.

The performance is not good enough, for example, in Tab.1, Ours (PseCo) only improves the baseline by 0.4%.

Thus this paper does not meet ICLR's standards.

**Questions:**

1. Consistent-Teacher is the best baseline in your paper, why don't you integrate your augmentation with it?
2. I would suggest solving the abovementioned question/weakness to make this paper solid before submission.

---

### Official Review · Reviewer_PUQb · 2024-11-04

**Soundness:** 2
**Presentation:** 1
**Contribution:** 2
**Rating:** 3
**Confidence:** 4

**Summary:**

The paper introduces a semi-supervised underwater object detection method(UIEG+) that  tries to address distribution differences between labeled and unlabeled images. The authors use 3D color memory and 1D scale memory to track image distributions, guiding transformations to ensure better detection.

**Strengths:**

Semi-supervised learning is important for underwater imaging problems due to high cost of annotation. UIEG+ is compatible with existing semi-supervised object detection (SSOD) frameworks and it introduces a unique method by considering both color and scale distributions.

**Weaknesses:**

Focusing on specific attribute like color and scale coulld be very limiting. The ablation study is good, but I'd prefer to see the mAP with only color transform.

The 3D color memory and 1D scale memory approach might introduce additional computational overhead. It mgiht be better to include some information about time-complexity.

It's not completely clear whether an image enhancement framework is needed for object detection. The mAP doesn't show constant improvement over existing methods.

**Questions:**

It's not completely clear whether an image enhancement framework is needed for underwater object detection or an end-to-end method would be better. It might be better to provide more detailed comparison with semi-supervised or self-supervised object detection methods.

It might be better to include some discussion about other attributes that could improve performance. More specifically edge/shape/frequency based attributes might be useful.

The results presented in the paper are quite close, making it difficult to assess the statistical significance of the improvements. Including error bars or confidence intervals would provide valuable insight into the robustness and reliability of these results.

---

### Official Review · Reviewer_AXLT · 2024-11-05

**Soundness:** 2
**Presentation:** 2
**Contribution:** 2
**Rating:** 3
**Confidence:** 5

**Summary:**

The authors present a paper on an image enhancement method for semi-supervised object detection in the underwater image domain. The proposed approach follows a teacher-student architecture, with the teacher initialized on labeled data and updated via EMA from the student, which is in turn trained on both labeled and pseudo-labeled images, the latter subject to the said augmentation approach. Experimental results on two datasets show that the method achieve performance at least on par with the state of the art.

**Strengths:**

1) The paper is well-written and easy to follow.

2) The proposed approach is able to achieve good results using a simple enhancement procedure.

**Weaknesses:**

1) The authors claim that the proposed approach takes into account the appearance distribution of unlabeled images, unlike other approaches from the state of the art. These approaches don’t take specifically into account the unlabeled distribution, but process all images in a uniform way; hence, if there is a main color/scale mode in the distribution, it is the one that will be mostly represented by the state-of-the-art approaches. However, this seems to achieve the same result as what the authors are doing, i.e., explicitly select the mode of the distribution.

2) Results lack confidence intervals or standard deviations, making it hard to assess the statistical significance of AP/mAP differences.

3) Overall, the methodological novelty of the approach is limited. The whole framework follows an established paradigm, and the enhancement approach is really very simple (besides my notes in weakness 1).

**Questions:**

1) How is weak augmentation performed?

2) The color and scale augmentations seem to always choose the most frequent bin in the corresponding memories. Why not employ a weighted sampling?

---

### Official Review · Reviewer_WyZx · 2024-11-08

**Soundness:** 2
**Presentation:** 3
**Contribution:** 2
**Rating:** 3
**Confidence:** 3

**Summary:**

In this work, the authors propose a novel underwater image enhancement method guided by attribute-based data distribution, which focuses on reducing the discrepancies between enhanced and original unlabeled images across different attributes, thereby effectively addressing the challenges in semi-supervised underwater object detection. Experimental evaluations were performed on multiple datasets, and the experimental results look good.

**Strengths:**

1. This paper proposed a novel underwater image enhancement method guided by attribute-based data distribution (UIEG+), which aims to reduce distributional differences between enhanced and unlabeled underwater images by analyzing the distribution of unlabeled images in terms of color and scale attributes.
2. This paper incorporate the proposed UIEG+ into existing SSOD frameworks, thereby effectively addressing the challenges of semi-supervised underwater object detection.

**Weaknesses:**

1. The experiment is not sufficient. The authors have discussed some recent related work in 2024, but did not compared with them.
2. The contribution is somewhat limited. A novel underwater image enhancement method guided by attribute-based data distribution (UIEG+) is proposed in the detection model. If using recent SOTA image enhancement instead in the detection model, will it improve the performance?
3. The ablation experiments are inadequate. For example, only CTransfor and STransfor components on URPC are tested.

**Questions:**

1. More recent work should be compared to verify the superiority of the proposed model.
2. The authors may discuss about the enhancement part, i.e., whether the SOTA enhancement methods can improve the performance or not. This can show the effectiveness of the contribution.
3. Please add more ablation experiments.

---

### Note · Authors · 2024-11-26

I have read and agree with the venue's withdrawal policy on behalf of myself and my co-authors.